# Experimental Study of the Bending Behaviour of the Neovius Porous Structure Made Additively from Aluminium Alloy

Katarina Monkova [1,2,*], Peter Pavol Monka [1,2], Milan Žaludek [2], Pavel Beňo [3], Romana Hricová [1] and Anna Šmeringaiová [1]

[1] Faculty of Manufacturing Technologies, Technical University in Kosice, Sturova 31, 080 01 Presov, Slovakia
[2] Faculty of Technology, Tomas Bata University in Zlin, Nam. T.G. Masaryka 275, 760 01 Zlin, Czech Republic
[3] Faculty of Technology, Technical University in Zvolen, Studentska 26, 960 01 Zvolen, Slovakia
* Correspondence: katarina.monkova@tuke.sk

**Abstract:** Porous materials bring components not only direct advantages in the form of lightening of constructions, saving of production materials, or improvement of physical properties, but also secondary advantages, which are manifested as a result of their daily use, e.g., in aviation and the automotive industry, which is manifested in saving fuel and, thus, environmental protection. The aim of this article is to examine the influence of the volume ratio of a complex porous structure, the so-called Neovius, on bending properties. Samples with five different relative weights of 15, 20, 25, 30, and 50% ($\pm1\%$) were fabricated from AlSi10Mg aluminum alloy by Direct Laser Metal Sintering (DLMS) technology. A three-point bending test until specimen failure was performed at ambient temperature on a Zwick/Roell 1456 universal testing machine. The dependences of the bending forces on the deflection were recorded. The maximum stresses, energy absorption, and ductility indexes were calculated to compare the bending behavior of beams filled with this type of complex cellular structure. The results showed that Neovius, with a relative weight of 50%, was much more brittle compared to the other samples, while the Neovius structure, with a relative weight of 30%, appeared to be the most suitable structure for bent components among those tested. This study is a contribution not only to the development of the space and aviation industry but also to the expansion of the knowledge base in the field of material sciences. This know-how can also provide a basis for defining boundary conditions in the simulation of behavior and numerical analyses of 3D-printed lightweight components.

**Keywords:** bending behaviour; cellular structure; relative weight; aluminium alloy; additive manufacturing

## 1. Introduction

The current development of additive technologies brings not only new possibilities but also new challenges. One of them is the use of cellular materials in various components and structures in order to fully utilize the potential of porous structures and their advantages. These benefits are related to predicting their behavior, weight reduction, energy absorption capacity, and material savings while maintaining the required safety and operational reliability of devices containing such components. The above requirements are often given to the parts used in the aerospace and aeronautical (as well as automotive) industries, as lightening the products undoubtedly brings benefits in fuel economy. Subsequently, this is reflected not only in saving operating costs and materials for the production of components but also in the protection of the environment. It is, therefore, very important for engineers to know the properties of such lightened materials and their behavior under different types of loading before applying them to final components [1–3].

However, the inherent complexity of lightweight bioinspired structures leads to several problems when lattices (or complex walls) need to be designed or numerically simulated. The computational power needed to capture the overall component is extremely high [4].

Another problem of simulation and numerical analyses still remains the inhomogeneity of the material due to the effect of material layering in 3D printing, even if the effect is not as significant as, for example, in the case of FDM technology in DMLS and SLS technologies. The behavior of structures under load and their resulting mechanical properties are also influenced by other parameters, such as the termination of the structure at the edges (boundaries) of the component, i.e., whether it is the whole cells or only their parts. The setting of boundary conditions is very complicated in the case of complex structures and requires an experimental basis for their rigorous determination [5,6].

In many cases, a simple uniaxial tensile or compression test may not provide all the information needed to understand different aspects of a material's behavior. When a specimen bends or flexes, it is subjected to a complex combination of forces, including tension, compression, and shear. For this reason, bend testing is commonly used to evaluate the reaction of materials to realistic loading situations [7,8]. Flexural test data can be particularly useful when a material is to be used as a support structure. For example, a plastic chair needs to give support in many directions. While the legs are in compression when in use, the seat will need to withstand flexural forces applied by the person seated. Not only do manufacturers want to provide a product that can hold expected loads, but the material also needs to return to its original shape if any bending occurs [9].

There are a number of studies dealing with the flexural properties of commonly available simply shaped beams fully filled with material or sandwich-type materials, but flexural testing of porous, cellular structures of complex shape is still an underexplored area. Only a few studies are available regarding the evaluation of the bending behavior of porous materials. The following studies can be mentioned as an example.

Gullapali [10], in his research, studied the flexural behaviour of 2D cellular lattice Structures Manufactured by Fused Deposition Modelling. They studied rectangular beams of lattice structures with different unit cell configurations, which were fabricated by the Fused Deposition Modelling (FDM) technique. All the specimens were fabricated on Stratasys Dimension 1200 ES FDM machine in Acrylonitrile Butadiene Styrene (ABS) material. Three-point bending tests were carried out to study flexural properties such as flexural strength and modulus of these cellular lattice structures. Results of flexural tests indicated that the cellular lattice structures based on triangular and honeycomb shapes exhibit maximum flexural strength.

The same FDM production technique and also ABS plastic material was used by Monkova et al. [11] for the bending properties investigation of a simple lattice structure based on a BCC (Body Centred Cube) cell. Cylindrically shaped samples were designed for testing and to compare their properties with a tube fully filled with ABS material. The results showed that a unit of material at this BCC type of lattice structure is able to carry down less stress compared to the sample fully filled by the material.

The nonlinear bending response of functionally graded porous beams reinforced by graphene platelets (GPLs) with various boundary conditions using the Ritz method was the research topic of Hung et al. [12]. They dealt with the nonlinear bending response of functionally graded porous beams reinforced by graphene platelets (GPLs) with various boundary conditions using the Ritz method. In the study based on the trigonometric shear deformation beam theory and the von Kárman type of geometrical nonlinearity strains, the system of nonlinear governing equations was derived using the minimum total potential energy principle.

Choi et al. [13] investigated samples fabricated by using ABS material and the structures constructed by the well-known honeycomb models using an FDM-Type 3D printer. To analyze the fracture surface of the samples constructed uniquely by using the 3D printer, the bending loads were applied to the samples at 30, $-10$, and $-50\,^\circ\text{C}$, respectively. From this experiment, it was evaluated that the fractured surface of the 3D sample was very rough at $30\,^\circ\text{C}$, while it was smooth at a relatively low temperature.

The elastic buckling and static bending analysis of shear deformable functionally graded (FG) porous beams based on the Timoshenko beam theory was a goal of Chen et al.,

while the elasticity moduli and mass density of porous composites were assumed to be graded in the thickness direction according to two different distribution patterns [14]. Similarly, Şimşek and Yurtcu [15], or Phuong et al. [16] used the nonlocal Timoshenko and Euler–Bernoulli beam theory to examine the analytical solutions for the static bending and buckling of an FG nanobeam.

Perhaps a few more studies could be found in connection with testing the bending properties of porous materials, but to the best of the authors' knowledge, the flexural properties of additively cellular materials with complexly shaped cells fabricated by the additive approach from aluminum alloys have not yet been addressed by researchers or only to a very small extent. Therefore, the authors of the presented article consider the combination of these elements in one study as a novelty and their contribution not only to the development of the space and aviation industry but also to the expansion of the knowledge base in the field of material sciences. This know-how can also provide a basis for defining boundary conditions in numerical analyses, as well as for the application of such structures in other branches of industrial economy or biomedicine.

The aim of this paper is, therefore, to investigate the bending properties of Neovius cellular structures produced with five different weight ratios of 15, 20, 25, 30, and 50% (±1%) from aluminum alloy AlSi10Mg by Direct Laser Metal Sintering (DLMS) technology. The Neovius surface is a triply periodic minimal surface (TPMS) originally discovered by Finnish mathematician Edvard Rudolf Neovius, whose base surface, reproduced opposite, has 12 holes centered on the edges of the cube, i.e., at the vertices of the cuboctahedron. The complete Neovius surface divides space into connected components, mutually isometric.

## 2. Materials and Methods

### 2.1. Samples Design

The behavior of lightweight cellular bodies in their operational conditions is given by a combination of the properties of the topology of cellular structure and the properties of the material used for its production. The main role also plays relative weight $W_R$ that expresses in percentages the ratio of the weight of the porous part (structure) in the body in relation to the weight of the same body that would be completely filled with the material, and it is defined by Equation (1) [17].

$$W_R = \frac{W_S}{W_{CFM}} \tag{1}$$

where
- $W_S$ is the weight of the porous structure in the body;
- $W_{CFM}$ is the weight of the body filled completely with the material.

For the presented research, the sandwich-type samples with the Neovius were selected as an interesting type of cellular structure, which has so far been rarely used but whose properties should be explored more deeply. The authors have already investigated the properties of this structure under tensile and compressive stress, while the results of these comparative studies indicate good mechanical properties of this structure regarding the volume ratio of the material, and they are being prepared for publication. In addition, Park et al. [18] claim in their study that the tensile properties of Neovius are similar to those of real bone. The basic cell of the structure and the complete Neovius surface are introduced in Figure 1.

The structure belongs to the group of so-called triply periodic minimal surfaces (TPMS). TPMS are the surfaces that locally minimize the surface area of the designated boundary so that the average curvature at each point on the surface is zero and can be further categorized into ligament/skeletal/solid-networks or sheet-networks. It can be mathematically described by Equation (2) [19].

$$f(x, y, z) = 3(\cos(x) + \cos(y) + \cos(z)) + 4\cos(x)\cos(y)\cos(z) \tag{2}$$

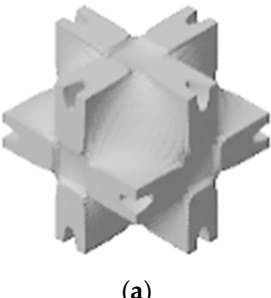
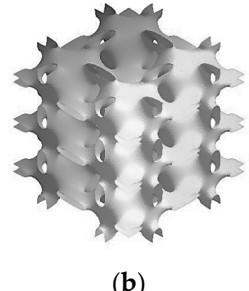

(**a**)                                                     (**b**)

**Figure 1.** (**a**) The basic cell of the Neovius structure and (**b**) the complete Neovious surface.

For this research, the virtual models of the samples with five different relative weights of 15, 20, 25, 30 and 50% ($\pm$1%) were generated in Rhinoceros 7 software with an additional Grasshopper tool. The main advantage of this CAD software is the ease of creating new structures using a block diagram that has been designed and modified according to the required research needs as well as to control relative via the wall thickness of the structure. On the other hand, this software allows the creation of any structures according to the entered mathematical equation, but without any major possible modifications, since the result of the creation of the structure is only a copied cell in the specified directions in the STL format, which no longer allows any significant modifications. Exporting the model in other formats is not possible. Moreover, it is not possible to perform finite element method (FEM) analysis and simulation on the model in this format.

The core of the sample with Neovius structure was created by multiplying the basic cell (sizes of 10 mm $\times$ 10 mm $\times$ 10 mm) in all three orthogonal directions *x*, *y*, *z*, while the number of cells in the directions was 3, 2, and 25, respectively.

The authors tried, within production possibilities as well as the testing capacities of the machine, to design samples in such a way that they were suitable for experiments. Samples with a volume ratio of less than 15% could not be produced, as the thickness of the wall did not allow the self-supporting function of the structure and, thus, the correct bonding of other layers of material to the previous layer. The self-supporting function was a condition due to the type of cell since, in the case of using support structures, it would be problematic, if not impossible, to remove these support structures after the 3D printing while maintaining a sufficient quality of the samples. The volume ratio of 15% was, therefore, the smallest possible from the point of view of manufacturing, and together with the following three volume ratios of 20, 25, and 30%, it was chosen from the point of view of the highest degree of weight lightening, which is the primary benefit of such structures. The highest selected volume ratio of 50% was considered by the authors to be borderline in terms of the weight-lightening function. At the same time, it was supposed to highlight the trend of the influence of the volume ratio on the properties of the samples in bending if the properties (stress, energy, or ductility) at four lower regularly distributed volume ratios 15, 20, 25, and 30% appeared to be functionally dependent. The number of cells was chosen considering the capabilities of the test equipment (distance of supports and capacity of the machine), while preliminary calculations were carried out with an additional number of cells at the maximum selected volume ratio $Vr = 50\%$ and also an experimental test of one preliminary sample was carried out. With these facts in mind, the number of cells $25 \times 3 \times 2$ was chosen, which at least partially considers the reproduction of the basic cell in the volume of the future component and "takes into account" the boundaries between cells. Given that it is a comparative study, the results should sufficiently interpret the behavior of the Neovius structure in bending and point to the influence of the volume ratio.

To prevent the cell wall from being deformed by the pressure of the thumb of the testing machine directly on the structure, it was covered with a continuous layer of material with a thickness of 1 mm. The total dimensions of the samples were, thus, 30 $\times$ 22 $\times$ 250 mm. The layout of the structure in the cross-sectional area of the sample is shown in Figure 2.

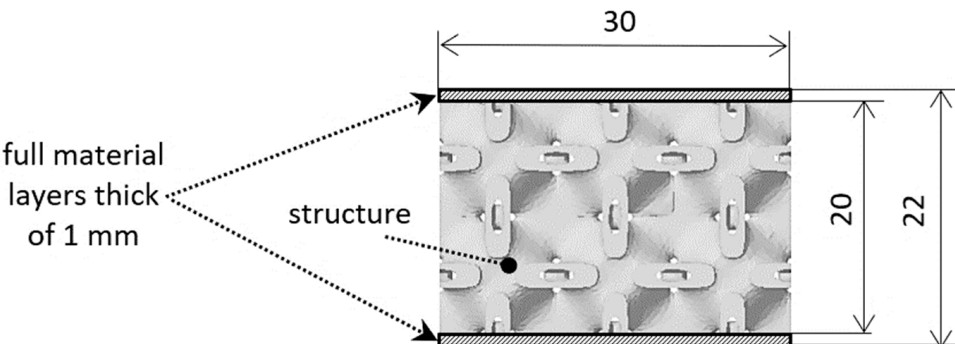

**Figure 2.** The cross-sectional layout of the sample design.

*2.2. Samples Production*

For the production of samples, the material of aluminium alloy AlSi10Mg (EU: 3.2381, USA: A 360), supplied for 3D printing in the form of powder, was chosen, which is suitable for a wide range of applications not only in the aviation and aerospace industry but also provides good applicability to other industries such as the automotive industry or engineering practice. It is a light structural alloy of lower density than other materials for 3D printing that shows good alloying properties, heat, and electric conductivity. The basic mechanical properties given by the powder manufacturer after a solid body production are as follows [20]:

- Ultimate tensile strength Rm = 410 MPa;
- Yield strength $Rp_{0.2}$ = 240 MPa;
- Young's modulus E = 70 $\pm$ 5 GPa;
- Elongation at break (as built) 5 $\pm$ 2%.

The chemical composition of the alloy is shown in Table 1 [20].

**Table 1.** Chemical composition of AlSi10Mg alloy [20].

| Element | Al | Mg | Si | Ni | Sn | Pb | Cu | Zn | Ti | Mn | Fe |
|---------|------|-----------|--------|-------|-------|-------|-------|------|-------|-------|-------|
| wt. (%) | Balance | 0.2 ÷ 0.45 | 9 ÷ 11 | <0.05 | <0.05 | <0.05 | <0.05 | <0.1 | <0.15 | <0.45 | <0.55 |

The samples were produced by Direct Metal Laser Sintering (DMLS) technology employing a 3D printer EP-M250 (with 500 W laser power with a focused diameter of 0.1 mm, scanning speed of 8 m/s, hatching distance of 0.15 mm, Ar gas supply, and a flow rate of 40 m/s). They were attached to the building platform by means of supporting structures and were produced with a layer thickness of 30 μm. Supplier of samples with long-term experience in the area of 3D printing of metal components and with DMLS technology guaranteed the quality of the produced samples, while the process parameters were determined not only as recommended by the 3D printing machine manufacturer but also according to its experience, as optimization of the parameters is closely related to the quality of the components manufactured and is important for the efficiency and sustainability of production [21–23]. The direction of the building corresponded to the vertical axis of the 3D printing machine workspace, while the continuous layer of the sample (completely filled with the material) was perpendicular to the building platform, while the *z*-axis corresponded to the direction of the sample building by adding individual layers, as is presented in Figure 3.

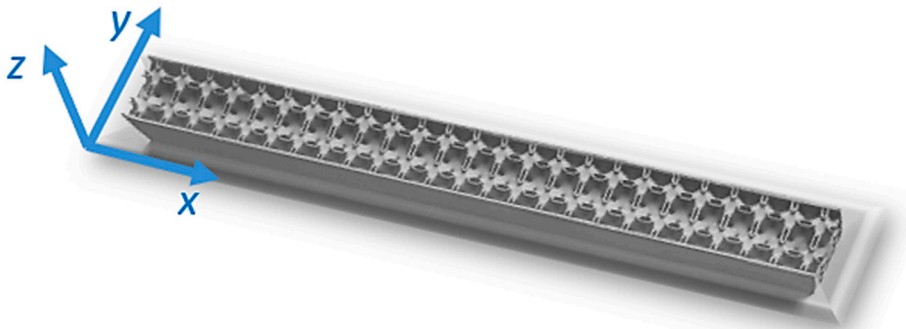

**Figure 3.** Orientation of the sample in the workspace of the 3D printer.

Additional processing consisted of heat treatment according to the recommendations of the powder material manufacturer [20] (heating up to 240 °C in 1 h with a speed of 8 °C/min, keeping for 5 h, and then air cooling), followed by removing from the platform and sandblasting. Three samples of the same type because of the test repeatability were produced. Due to the repeatability of the test, three samples of the same topology (type of structure and weight percentage) were produced. An example of a set of samples (five different relative weights) is shown in Figure 4.

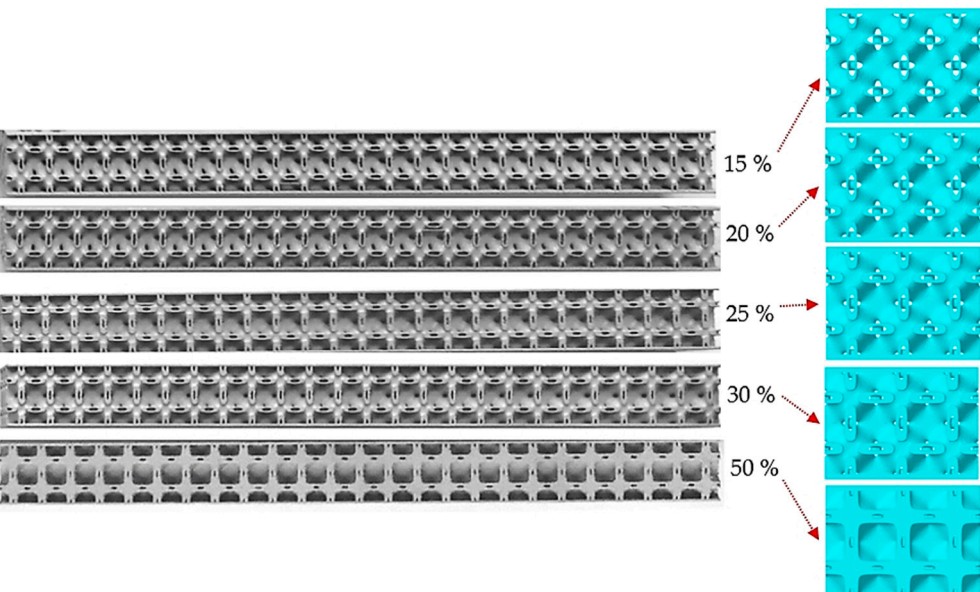

**Figure 4.** The cross-sectional layout of the sample design.

### 2.3. Experimental and Evaluation Methodology

2.3.1. Experimental Procedure

Tensile testing cannot be used to determine bendability since these are different failure modes. Failure in bending is similar to other modes limited by local formability in that only the outermost surface must exceed the failure criteria. ASTM E290A-26, ISO 7438I-8, and JIS Z2248J-5 are some of the general standards which describe the requirements for the bend-testing of metals. In a three-point bending test, a supported sample is loaded at the center point and bent to a predetermined angle or until the test sample fractures.

Flexural strength, also known as modulus of rupture, bend strength, or fracture strength, is a material property defined as the stress in a material before it yields in a flexure test. The flexural failure represents the highest stress experienced within the material at its moment of rupture. It is measured in terms of stress and mostly occurs at the mid-span of a beam [24]. Compared to ductile materials, brittle samples usually show different behavior in flexural tests, while the sample often breaks without any noticeable deformation [25].

Determining a flexural (offset) yield point is difficult for such materials. Therefore, in the case of brittle materials, it is not the onset of plastic deformation that is used as the material strength but the onset of fracture when the maximum bending moment $M_{max}$ is reached. This strength parameter is then called (ultimate) flexural strength, bending strength, or modulus of rupture [26,27].

To investigate the properties of complex porous structures, a three-point bending test up to the failure of the sample was used in the research. The experiments were carried out according to the ISO 7438:2020 standard at an ambient temperature of 22 °C and a humidity of 60% on a ZWICK 1456 testing machine with a force of 250 kN and a load cell head of 20 kN. The specimens were set centered on supports 220 mm apart such that a 5 mm rounded push thorn acted perpendicular to the continuous layer of material covering the specimen structure, and the crossbar moved at a feed of 20 mm/min. The testing machine with a detail of the sample placed in the test position is shown in Figure 5.

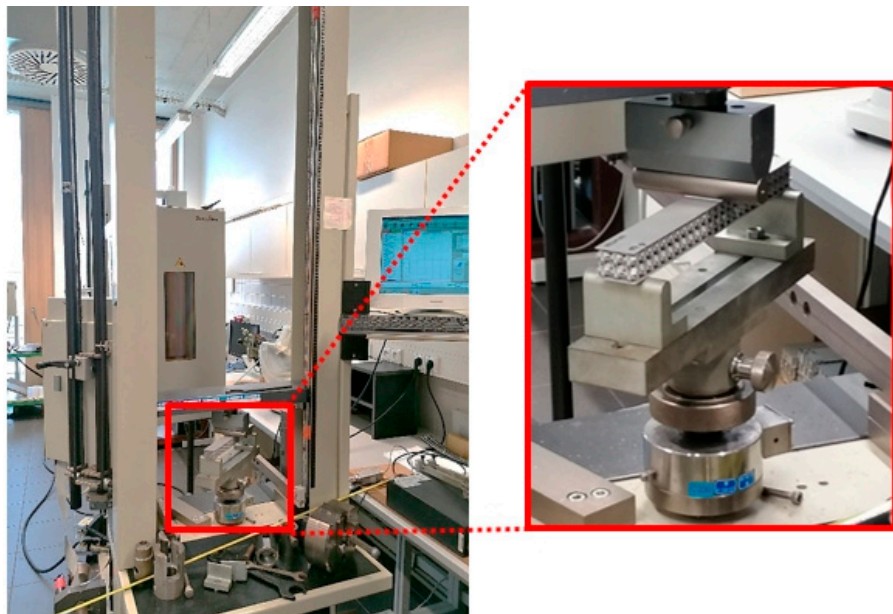

**Figure 5.** The testing machine ZWICK 1456 with a detail of the sample placed in the test position.

2.3.2. Maximum Bending Stress Calculation

The bending test is a destructive test used to determine the strength and ductility of materials. The strains of the beams in a middle profile were estimated based on the changes in the deflections of the measurement points in the *xy* plane. Within the research, a 3-point bending test was used (Figure 6). Given a particular beam section, it is obvious that the bending stress will be maximized by the distance from the neutral axis *y*. Thus, the maximum bending stress $\sigma_{max}$, given by Equation (3), will occur in the middle of the beam, either at the top or the bottom of the beam section, depending on which distance is larger, and it can be as follows:

$$\sigma_{max} = \frac{M_{max}y}{I_{xx}} \tag{3}$$

where

- $M_{max} = Fl/2$ is the maximal bending moment about the section's neutral axis (Nmm);
- *y* is the perpendicular distance from the neutral axis to the farthest point on the section (mm);
- *Ixx* is the second moment of area of the cross-section (moment of inertia) of the beam, about the neutral *x*-axis (mm$^4$).

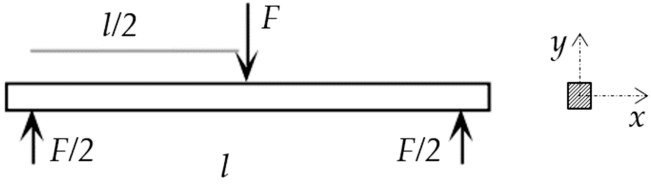

**Figure 6.** Three-point bending test.

Due to the complexity of the cellular structure, the *.stl files (as Rhinoceros 7 software outputs) were imported into PTC Creo software, where the facets were cleaned, and the software was used to determine the moment of inertia (Figure 7). The software confirmed the almost complete coincidence of the axes of the coordinate system located at the center of gravity of the beam with the principal stress axes.

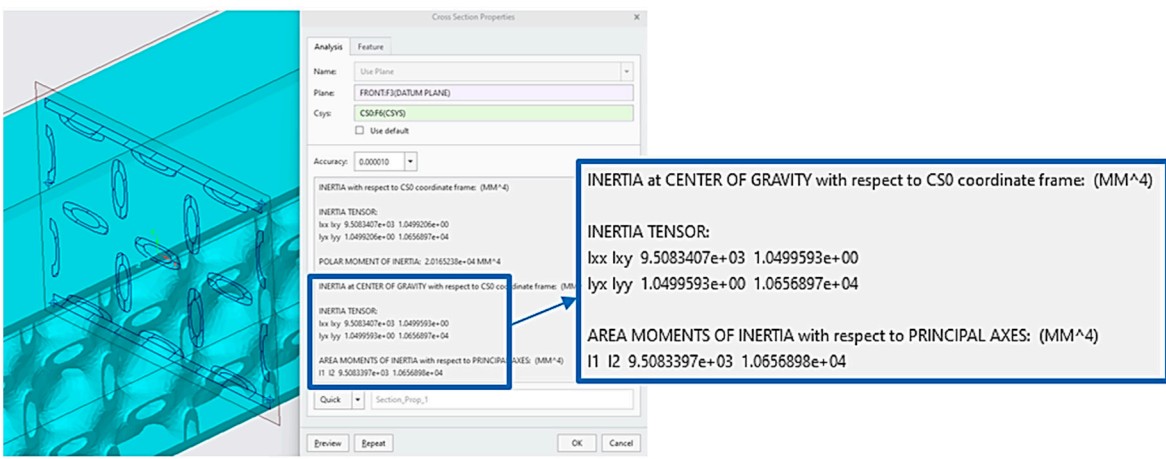

**Figure 7.** Moment of inertia specification.

Based on the dependence of the load on the deflection, the maximum force was determined, and then the maximum stresses were calculated.

### 2.3.3. Energy Absorption

To compute the energy absorption, the curve of dependence of load on deflection was expressed by a polynomial function. The area under the curve (see Figure 8), which corresponds to the magnitude of the absorbed energy of the beam during the test, was counted by integrating the load with respect to the deflection.

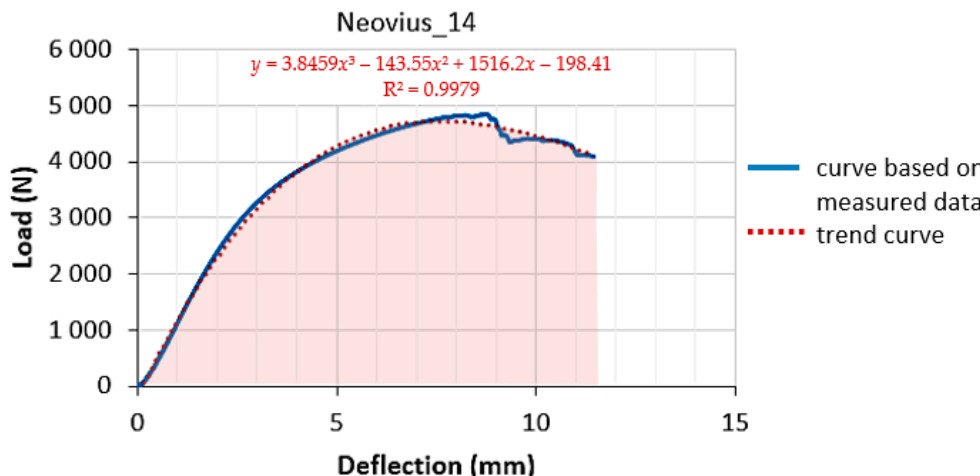

**Figure 8.** Calculation of energy absorption based on the force-deflection curve.

The measured data were transferred to MS Excel, where the dependences of the load on the deflection were plotted in a graphical way, in which a trend curve with a polynomial function was also displayed. The 3rd-order polynomial function was most often used.

### 2.3.4. Ductility

In the present research, the bending ductility of the beams was also evaluated. Bending ductility can be defined as the capability to withstand plastic deformations by maintaining the appropriate level of load-bearing capacity [28].

Two indices, $\mu_d$ and $\mu_E$, were used for the ductility of the beam assessment. The ductility index $\mu_d$ is based on the deflection value at the proportionality limit, and it can be described by Formula (4) [29,30]:

$$\mu_d = \frac{u_u}{u_e} \tag{4}$$

where

- $u_u$ is the deflection at the ultimate load (mm);
- $u_e$ is the deflection at the elastic limit (mm).

The ductility index $\mu_E$ is expressed as the quotient of the total and elastic energy and is given by Formula (5) [31,32].

$$\mu_E = \frac{1}{2}\left(\frac{W_{tot}}{W_e} + 1\right) \tag{5}$$

where

- $W_e$ is the elastic energy (fraction of total), the area under the load-deflection curve up to the elastic limit (J);
- $W_{tot}$—the total energy, the area under the load-deflection curve up to failure (J).

### 3. Results and Discussions

This section evaluates the bending behavior of Neovius 3D-printed cellular structures from different points of view. The designation of the samples consisted of two specifications: the type of structure and the relative weight (%) counted according to Equation (1).

Bending tests were performed until a specimen failed, as is presented in Figure 9, where a detail of crack propagation in the sample Neovius_20 is also shown.

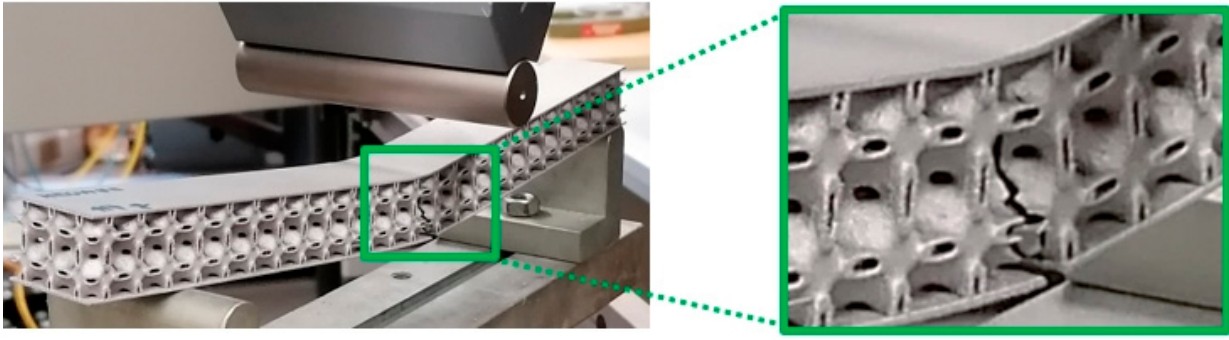

**Figure 9.** The sample after failure with detail of crack propagation.

Based on the experimental procedure, the curves of the dependence of the load on the deflection were plotted using the TestXpert software, which is part of the testing machine. An example of the representative curves obtained for one set of samples is in Figure 10.

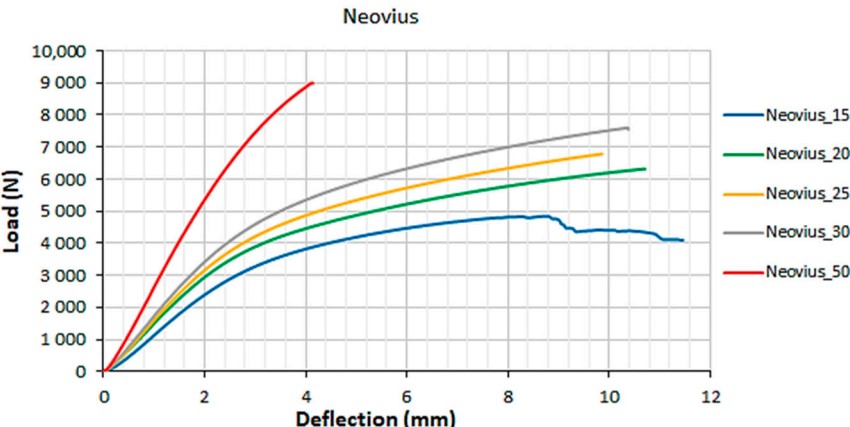

**Figure 10.** Load-deflection curves for 5 different relative weights of the Neovius structure.

From the dependences of the load on the deflection, it is clear that as the relative weight of the structure increases, so does the load required to fail the sample, which could be predicted. It is interesting that the deflection at failure for all three pieces of specimens with Neovius structures at a relative weight $W_R = 30\%$ was greater than for structures with $W_R = 25\%$.

Based on the dependencies, subsequently, maximum stresses were computed, and trend curves were generated in the MS Excel software application, which mathematically described the experimentally obtained curves very accurately (with the minimum coefficient of determination, $R^2 = 0.998$). By integrating the obtained polynomial functions, the amount of energy needed up to a failure of the sample was calculated for each sample. The values with standard deviations of the maximum stresses and energy absorptions for the samples characterized by individual relative weights are presented via the diagram in Figure 11.

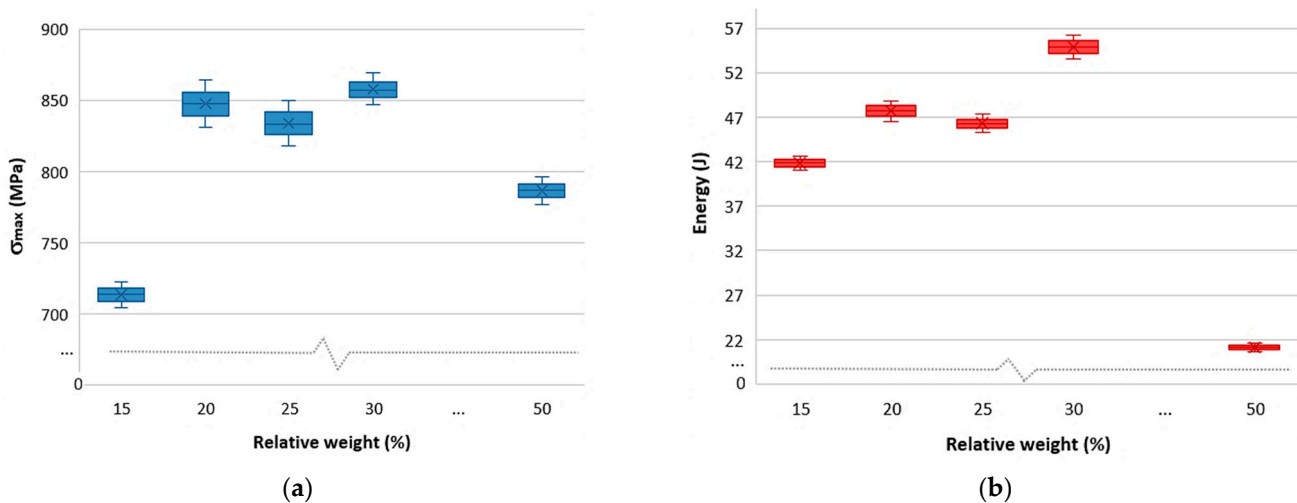

**Figure 11.** Measured values with standard deviations of (**a**) the maximum stress and (**b**) energy absorption for individual relative weights of the Neovius samples.

At first glance, it is clear that the values of the maximum stresses for the samples with relative weights 20, 25, and 30% do not differ significantly, while the standard deviations as the variance of the data set in relation to its mean indicated that the measured data are distributed relatively close to the mean, which can also indicate that the samples were produced in the same or very similar quality (without errors arising from the influence of technology). The difference was manifested in the Neovius_30 sample in the amount of absorbed energy, which clearly exceeded the values of the other samples.

The highest maximum stress of 857 MPa was shown by Neovius_30, although samples with 20 and 25% relative weight reached similar values, i.e., 847 and 833 MPa. The average maximum stress for samples with 50% relative weight was only slightly lower compared to samples with 20 and 25% relative weight (it was exactly 787 MPa), but from the graph in Figure 10b, it is clear that the Neovius_50 samples had the lowest energy absorption (with the value being significantly lower compared to other samples), from which it could be judged that the behavior of Neovius_50 was much more brittle than the other samples. Neovious_30 samples had the best energy absorption properties.

For a more comprehensive evaluation of the bending behavior of the samples, it was also necessary to take into account the assessment of ductility, which is mechanical property commonly described as a material's amenability to drawing. In bending, ductility is defined by the degree to which a material can sustain plastic deformation under bending stress before failure. It is an important consideration in engineering and manufacturing since it defines a material's suitability for certain operations and its capacity to absorb mechanical overload [33–36].

Ductility within the presented research was evaluated via two indexes (Figure 12), specifically by $\mu_d$ (based on the deflection value at the proportionality limit) and by $\mu_E$ (expressed as the quotient of the total and elastic energy).

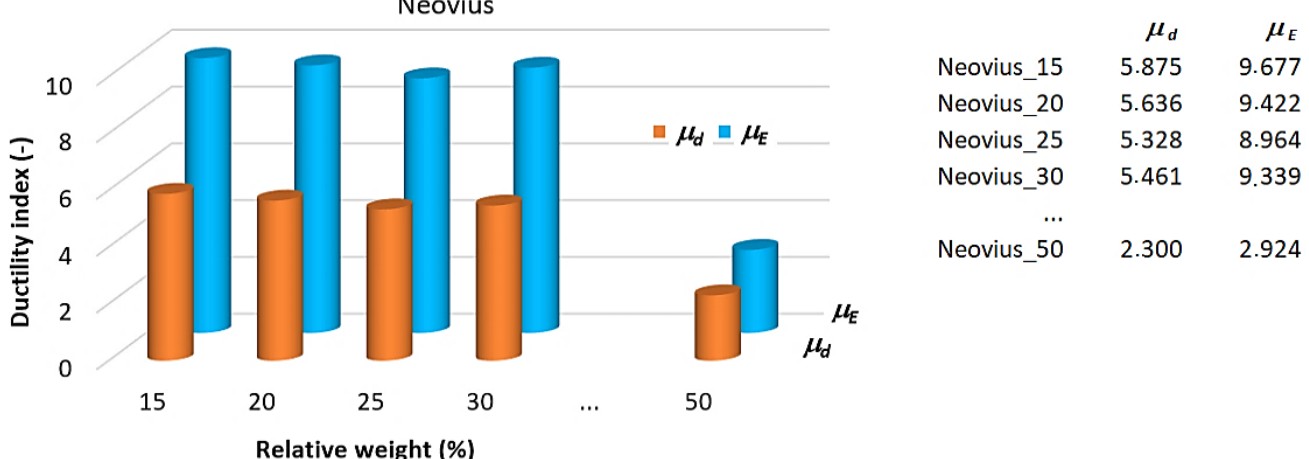

**Figure 12.** Ductility indexes evaluation.

It is clear from the histogram in Figure 12 that both ductility indexes $\mu_d$ and $\mu_E$ are very similar for the specimens with 15, 20, 25, and 30% relative weights, but for Neovius_50, they are significantly lower. It confirmed the much more brittle behavior of the structure Neovius with 50% relative weight compared to structures with a lower relative weight. The best ductility was shown by the Neovius_15 samples, and based on the evaluation of the experimental data, structures with a relative mass higher than 50% can be expected to behave brittle. The cracks at individual specimens are completed and visualized in Figure 13.

Flexural failure was identified in the way that almost only the transverse cracks developed during loading, whereas flexural–shear failure was only partially observed in a small number of cases where small diagonal cracks appeared, yet the structure failed due to the main transverse cracks. It could be said that the greatest susceptibility to brittle fracture showed the structure with the largest investigated volume ratio of 50%.

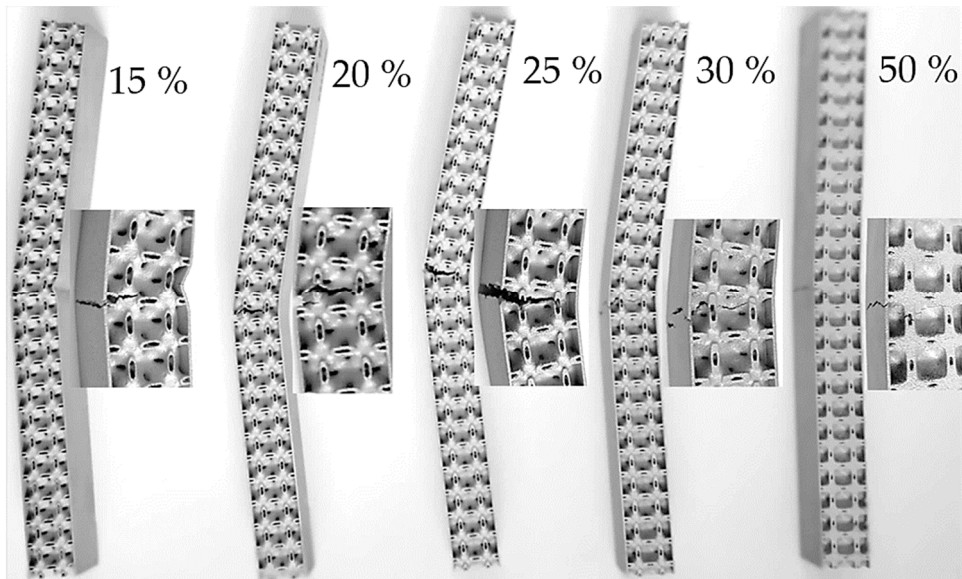

**Figure 13.** Visualization of cracks at individual specimens.

In regard to the primary benefit of the structure, which is lightening, it is possible to consider its use in the aerospace, aeronautical, automotive, engineering, or construction industries, and while taking into account all the data and evaluating the results, it is, therefore, possible to say that among the examined samples, the Neovius structure with a relative weight of 30% appears to be the most suitable choice for the components exposed to bending.

## 4. Conclusions

In nature, there are many types of materials with porous structures, which have become an inspiration for the development of new materials used in various sectors of industry, economy, and everyday use. Each of the porous structures, whether natural or artificially made, exhibits different properties. For a designer who wants to use this type of lightweight material in a design, it is, therefore, advantageous to know the behavior of individual types of structures and to be able to choose the one that is most suitable from the point of view of the expected stress on the component (product).

The aim of the article was to study the bending behavior of the complex triply periodic porous structure Neovius produced from AlSi10Mg alloy by DMLS technology.

- Within the presented research, the maximum bending forces of the samples (a sandwich type) with five different volume ratios, 15, 20, 25, 30, and 50%, were measured, as well as the dependences of force on deflection were plotted.
- The maximum stresses and the amount of energy consumption were calculated for individual specimens that differed in relative weight. The results indicated much more brittle behavior of the specimens with 50% relative weight in comparison to others. It was also confirmed by ductility evaluation.
- From the investigated samples, the most suitable choice for an application in aerospace, aviation, automotive, mechanical, or civil engineering practice for components subjected to bending appears to be the structure Neovius, with a relative weight of 30% due to the reported properties concerning the amount of material spent in production (and its weight).
- At the conclusion of the presented research, it can be stated that the achieved results can be considered as a basis for the design of the various components (parts of different machines, equipment, vehicles, and handling means), which are expected to be stressed predominantly (statically and dynamically) by bending. The implementation of such components in the technical equipment will not only reduce the total weight

of the system but also save the material needed for the production of such a component, as well as the mechanical properties determined by the designer with regard to product quality, safety, and operational reliability, will be preserved.

The study expands knowledge in the field of materials science and engineering, as, to the best of the authors' knowledge, the behavior of the Neovius structure made by DMLS technology from aluminum alloy under bending loading and taking into account the influence of the relative weight of the structure has not yet been published. In the near future, the authors would like to investigate the behavior of other types of cell structures under different types of loads.

**Author Contributions:** Conceptualization, K.M. and P.P.M.; methodology, K.M. and P.P.M.; software, K.M. and P.P.M.; validation, P.P.M. and P.B.; formal analysis, P.B., A.Š. and R.H.; investigation, M.Ž. and K.M.; resources, R.H., A.Š. and P.B.; data curation, K.M., M.Ž. and P.B.; writing—original draft preparation, K.M.; writing—review and editing, P.P.M. and R.H.; visualization, K.M., P.P.M. and A.Š.; supervision, K.M.; project administration, K.M.; funding acquisition, K.M. All authors have read and agreed to the published version of the manuscript.

**Funding:** This research was funded by the Ministry of education, science, research and sport of the Slovak Republic, grants APVV-19-0550, KEGA 005TUKE-4/2021, KEGA 032TUKE-4/2022, and SK-CN-21-0046.

**Data Availability Statement:** Not applicable.

**Acknowledgments:** The article was prepared thanks to the support of the Ministry of Education of the Slovak Republic through the grants APVV-19-0550, KEGA 005TUKE-4/2021, KEGA 032TUKE-4/2022 and SK-CN-21-0046.

**Conflicts of Interest:** The authors declare no conflict of interest.

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
