# Peer review of "Experimental Study of the Bending Behaviour of the Neovius Porous Structure Made Additively from Aluminium Alloy"

_aerospace, doi:10.3390/aerospace10040361_

Round 1

Reviewer 1 Report

The current manuscript investigates the bending behaviour of a specific additively manufactured structure. The title is attractive and is highly recommended for applying additive manufacturing research for direct applications. However, some significant issues should be considered as follows:

- The abstract needs to be carefully revised focusing on the outline of importance, applications, problem statement, and main contributions and novelty of the current work. 

- The introduction lacks a critical review of metal additive manufacturing. The defects in the fabricated parts, the current challenges and the literature efforts should be presented as well as the advantages. 

- There is no need to add a review about the FDM technology which is related to only plastics. The focus should be related to metals as it is the applied material in the current research. 

- The problem statement should be well defined in the introduction section.

- The materials and methods section should present powder characterization, the additive manufacturing machine used, and the process parameters.

- The applied heat treatment procedures should be related to a reference.

 - What is the element type of the lattice structure? and why is it selected? the answers should be clear in the manuscript text.  

- Figure 3 and 4 do not add meaningful data, a scale should be applied to the images and a zoom-in view should illustrate the difference between the applied relative densities.  

- A deep characterization is highly recommended to justify the obtained and claimed results of the bending behaviour. Microstructure characterization is recommended.

- The results and discussion section lacks references to support the presented results and analysis.

- the conclusion section should focus on including the main results, contributions and novelty of the current work.

Author Response

Dear Reviewer!

Thank You very much for Your valuable comments. We received them with great respect. We also would like to thank You for the possibility to make the improvements. We appreciate it very much.

All changes in the manuscript are highlighted in green, while our responses are coloured blue in the enclosed document "Responses to the Reviewer 1 comments"

At the same time, we apologize very much because we mistakenly uploaded the reactions to another reviewer's comments, so we added the second one. Please, consider the file named "Responses to the Reviewer 1 Comments DEF" as relevant for You.

Reviewer 2 Report

The comments and suggestions to improve the overall quality of the manuscript are contained in the attached pdf file.

Author Response

Dear Reviewer!

Thank You very much for Your valuable comments. We received them with great respect. We also would like to thank You for the possibility to make the improvements. We appreciate it very much.

All changes in the manuscript are highlighted in green, while our responses are coloured blue in the enclosed document.

Reviewer 3 Report

1.      The literature review section introduces many references of plastic 3D printing materials. How do those cited papers motivate the authors to study the metal Neovius structures?

2.      The paper indicates that Neovius structure properties could be promising. If that is the case, why is this structure rarely used?

3.       Is “cantered” a typo?

4.      The unit cell is a cubic GBV with constant size 10x10x10 mm. What parameters of Neovius unit cell are changed to achieve 5 different relative weights?

5.      How is “flexural failure” defined? It is better to elaborate on it in the context.

6.      What is the standard deviation of max stress? Is there a statistical computation showing the max stress of Neovius_30, Neovius_25, and Neovius_20 samples are “significantly” different?

7.      In conclusion, it is pointed out that volume ratios of 10, 15, 20, 25 and 35 % were measured. This contradicts with the prior sections.

8.      Relative weight 30 % is reported to be the most suitable choice. This could be biased because there are no testing results regarding 35% or 40% relative weight to compare with. 

Author Response

(The authors gave the same response as above.)

Round 2

Reviewer 1 Report

The revised manuscript is improved and most of the review comments and recommendations are well addressed. However, some issues are still need to be considered as follows:

- The authors should include the applied DMLS process parameters, there is no need to present the capabilities of the used printers. 

Some references could support the impact of optimizing the process-parameters to the quality of fabricated components;

Aboulkhair, Nesma T., et al. "3D printing of Aluminium alloys: Additive Manufacturing of Aluminium alloys using selective laser melting." Progress in materials science 106 (2019): 100578. 

- The presented results and analysis should be justified using independent characterization methods; for example the micro-hardness test should validate the claim for the ductility behavior. 

- The authors stated in the conclusion that "The study is contribution not only to the development of the space and aviation industry but also to the expansion of the knowledge base in the field of material sciences. This know-how can also provide a basis for defining boundary conditions in the simulation of behaviour and numerical analyses of 3D-printed lightweight components.". There is no evidence in the current study to consider that concept, however, this can be achieved after developing a simulation model, in addition to presenting a full characterization of the fabricated samples. 

One more review round is highly recommended after considering the above issues.

Author Response

Response to Reviewer´s 1 comments:

Dear Reviewer!

Thank You very much for Your valuable comments. We received them with great respect. We also would like to thank You for the possibility to make the improvements. We appreciate it very much.

All changes in the manuscript and responses are highlighted in blue.

Reviewer: The revised manuscript is improved and most of the review comments and recommendations are well addressed. However, some issues are still need to be considered as follows:

  • Reviewer: The authors should include the applied DMLS process parameters, there is no need to present the capabilities of the used printers. 

The parameters of the DMLS process are very closely related to the capabilities of the printers, and only research centres are allowed to change the parameters given by the machine manufacturer. Manufacturing companies adhere to the specified parameters (even if their change is allowed) since the machine manufacturer does not guarantee repair in the event of a malfunction of a machine intended for printing with DMLS technology, as these are very expensive 3D printers. Even changing the material in the form of powder for another within one machine is very difficult, and when changing the material it is not only necessary to clean everything, but this change requires the work of workers with high security (protective clothing and protective face mask).

However, in addition to the already mentioned parameters in the original post, we managed to find out other process parameters that were used in 3D printing. They are:

 “500 W laser power with a focused diameter 0.1 mm, scanning speed 8 m/s and hatching distance 0.15 mm, Ar Gas Supply, a flow rate of 40 m/s, a layer thickness of 30 µm.”

If any additional information about the process parameters needs to be specified, please let us know exactly what they are and we will ask the sample manufacturer.

  • Reviewer: Some references could support the impact of optimizing the process-parameters to the quality of fabricated components;

We thank the Reviewer for his comment. Until now, within the framework of the state of the art, we did not focus on the optimization of process parameters, as the presented research was not focused on the production technology, but on the mechanical properties of the produced porous samples. Of course, we recognize that the mechanical properties of the samples are closely related to their quality, but the optimization or change of the parameters was not within our capabilities, since, as it is mentioned in the manuscript, the samples were made to order and the supplier with a long term experience in3D metal printing guarantees their quality (otherwise he risks his name and reputation).

To point out the connection between the technology and the properties of the product, as well as the importance of optimizing the production process for the given conditions, we have argued the following references to the relevant part of the manuscript

  1. Krishnan M. et al., On the effect of process parameters on properties of AlSi10Mg parts produced by DMLS, Rapid Prototyping Journal, 20/6, 2014, 449–458, DOI 10.1108/RPJ-03-2013-0028
  2. Aboulkhair, Nesma T., et al. 3D printing of Aluminium alloys: Additive Manufacturing of Aluminium alloys using selective laser melting." Progress in materials science 106 (2019): 100578.‏‏
  3. R K Shah and P P Dey: Process parameter optimization of DMLS process to produce AlSi10Mg components, 2019, J. Phys.: Conf. Ser., 1240 012011, doi:10.1088/1742-6596/1240/1/012011

  • Reviewer: The presented results and analysis should be justified using independent characterization methods; for example the micro-hardness test should validate the claim for the ductility behavior. 

We thank the Reviewer for his opinion. We partially agree with him. However, we have to admit that despite many attempts in several types of software, we still have not been able to create a suitable network for analysis, as the sample consists of almost a million polygons and software in which 3D samples were geometrically defined allows you to export data only in *STL file, What is enough for production, but it is a big problem for numerical analysis, as there are many mistakes in the network and repair requires time and patience. On the other hand, the numeric model, if properly built, should correspond to the experimental testing, which is a priority in evaluating the results.

At the same time, we admit that microscopic analysis of failure areas has not yet been completed in this part of this investigation and will be the next step after the collecting results of experimental testing of other types of samples.

If Reviewer and Editor agree, we propose a change in the title of the article so that it is clear that the presented study was aimed at experimental testing of NEOVIUS-type samples (with different relative weights) made by additive access from aluminium alloy. The new title would be

„Experimental study of the bending behaviour of the Neovius porous structure made additively from aluminium alloy”

  • Reviewer: The authors stated in the conclusion that "The study is contribution not only to the development of the space and aviation industry but also to the expansion of the knowledge base in the field of material sciences. This know-how can also provide a basis for defining boundary conditions in the simulation of behaviour and numerical analyses of 3D-printed lightweight components.". There is no evidence in the current study to consider that concept, however, this can be achieved after developing a simulation model, in addition to presenting a full characterization of the fabricated samples. 

We thank the Reviewer also for this his opinion, which we respect and accept.

However, the authors consider their statement to be still valid, since, as mentioned above, experimental research is always a priority and the boundary conditions are set and defined within the new research area in the software according to the experimental results (of course,  realized by a sufficient number of experiments under the same conditions to confirm their repeatability, as well as to eliminate measurement uncertainties). Since the behaviour of complex structures of different types made of different materials with different topologies, material volume fractions, or porosity is still not a sufficiently researched area, we consider this experimental research as a contribution to the correct definition of boundary conditions in numerical simulations. Many studies deal with the numerical analysis of a single cell, but in real practice, lightweight components are formed by distributing and patterning basic cells. But it is precisely when cells are multiplied and powered that problems arise when powering network nodes and elements. Repairing such networks is very difficult and time-consuming. However, we believe that the problem of creating networks in such structures with complex geometry will be solved and numerical analyses will be performed soon.

Reviewer 3 Report

No further comments. 

Author Response

Dear Reviewer!
Thank YOU very much for YOUR words and for YOUR positive opinion!
We appreciate it very much.

Yours sincerely,

          Katarina Monkova (on behalf of the authors)

Round 3

Reviewer 1 Report

Most of the review comments, questions, and recommendations are well addressed. However, some issues still need to be considered as follows:

- The direct metal laser sintering for AlSi10Mg need to be confirmed and revised as the applied laser power of 500W and scan speed of 8000 mm/s are relatively too high compared to their values in the literature. 

- The discussion should compare the current study bending behaviour to those were published in the literature studies.

- The direct applications of the proposed structure and the applied material should be included in the conclusion section. 

Author Response

Responses to the Reviewer_3rd round

Reviewer: Most of the review comments, questions, and recommendations are well addressed. However, some issues still need to be considered as follows:

We thank the Reviewer for his opinion and comments. We appreciate his care to get the manuscript to a higher level.

  • Reviewer: The direct metal laser sintering for AlSi10Mg need to be confirmed and revised as the applied laser power of 500W and scan speed of 8000 mm/s are relatively too high compared to their values in the literature. 

We would like to reiterate that the article is not focused on the impact of processing parameters on the quality of 3D printing. These settings for samples made on the order were in the competence of the company since none of the co-authors´ academic workplaces doesn´t own such expensive equipment (and moreover, conditions for research at the faculties of the interested co-authors do not enable purchasing of the machine for DMLS technology and its maintenance so that it would be sustainable or profitable for a faculty just for the research activities). Despite the fact that the responsibility for the quality of the samples produced in this case was taken over by the company, all circumstances (from the size of the submitted files with CAD models of samples and quality of virtual models, up to the resulting quality of physically produced samples) were discussed with the manufacturer in many e-mails and as evidence, we are attaching photos from e-mail correspondence, where the quality of various types of samples and the impact of procedural parameters on their production was discussed. (Note: Samples with NEOVIUS structure for the bend test were ordered for production along with other types of samples for different types of experimental tests with different types of loads).

The pictures are included in the attached file.

The basic settings of processing parameters have already been included in our original manuscript (submitted almost 1 month ago) as we received them from the manufacturer after discussions and after setting up the most suitable conditions for their quality production. Influencing these settings was not in our competence.

  • Reviewer: The discussion should compare the current study bending behaviour to those were published in the literature studies.

As mentioned in the original article and also in the following reactions to the comments of reviewers, according to the best knowledge of authors, similar research in the behaviour of samples produced by DMLS technology from aluminium alloy when during bending has not been published yet, what the authors consider as a novelty, ... So, it is very difficult to compare the results obtained by other scientists under other conditions (other materials, structure, ...). They can be discussed and considered as the basis for our research, which we have done in the introductory part within the state-of-the-art (but they can not be directly compared).

  • Reviewer: The direct applications of the proposed structure and the applied material should be included in the conclusion section. 

Research submitted for assessment within the manuscript is part of the project of fundamental research. The aim of the project is to get to know the behaviour of different types of meta-structures with different porosity made of different materials so that it is possible to assess which combination of parameters is best suited for a particular type of load/stress. The presented experiments will be followed by tests of the fatigue load in bending, (also other methods of static or dynamic testing of samples with different types of structure are in the process of realization or will follow in the short future within the project). Once the characteristics of individual types of structures are detected, it will be possible to use those with the most suitable properties for the expected type of load within the components that are applied in the framework of technical practice, aerospace, aviation, automotive or everyday usage and to achieve reducing their weights while to maintain their functional properties.

The following statement was added to the manuscript:

The results of the research will therefore be applied to various components (parts of different machines, equipment, vehicles and handling means), which are expected to be stressed predominantly (statically and dynamically) by bending.
